# Syntenin: PDZ Protein Regulating Signaling Pathways and Cellular Functions

**DOI:** 10.3390/ijms20174171

**Published:** 2019-08-26

**Authors:** Tadayuki Shimada, Shin Yasuda, Hiroko Sugiura, Kanato Yamagata

**Affiliations:** Synaptic Plasticity Project, Tokyo Metropolitan Institute of Medical Science, 2-1-6 Kamikitazawa, Setagaya-ku, Tokyo 156-8506, Japan

**Keywords:** syntenin, membrane architecture, synapse, tumor metastasis, exosome biogenesis

## Abstract

Syntenin is an adaptor-like molecule that has two adjacent tandem postsynaptic density protein 95/Discs large protein/Zonula occludens 1 (PDZ) domains. The PDZ domains of syntenin recognize multiple peptide motifs with low to moderate affinity. Many reports have indicated interactions between syntenin and a plethora of proteins. Through interactions with various proteins, syntenin regulates the architecture of the cell membrane. As a result, increases in syntenin levels induce the metastasis of tumor cells, protrusion along the neurite in neuronal cells, and exosome biogenesis in various cell types. Here, we review the updated data that support various roles for syntenin in the regulation of neuronal synapses, tumor cell invasion, and exosome control.

## 1. Introduction

Syntenin was first isolated as an interacting partner of the syndecans, cell surface signaling, and trafficking proteins [1]. Simultaneously, syntenin was identified as a melanoma differentiation-associated gene product, namely, mda-9, whose expression was upregulated by interferon-γ treatment [2]. Syntenin is an ~32 kDa protein that contains two postsynaptic density protein 95/Discs large protein/Zonula occludens 1 (PDZ) domains. The PDZ domains were first identified as regions of sequence homology found in diverse signaling proteins [3,4]. The name PDZ derives from the first three proteins in which these domains were identified: PSD-95 (Postsynaptic density protein 95), Dlg (Discs large protein), and ZO-1 (Zonula occludens 1). The PDZ domain is a conserved sequence element and a well-characterized region comprised of six β strands and two α helices. PDZ domains bind to the C-terminus of the partner protein and are differentiated by the peptide-binding specificity, suggesting that syntenin works as an adaptor protein.

Syntenin has been shown to interact with a variety of proteins [5] and with the phospholipid, phosphatidylinositol 4,5-bisphosphate (PIP_2_) [6]. In its role as a multifunctional adaptor protein, syntenin has been implicated in a number of cellular events, such as receptor protein trafficking [7], developmental regulation of neuronal membrane architecture [8,9], synapse formation [10], axonal elongation [11], exosome production [12], and even the pathogenesis of various types of cancer cells, especially in invasion and stemness regulation [13,14]. Thus, syntenin mainly interacts with membrane-bound proteins and is involved in the subcellular trafficking of binding partners during endocytic and exocytic events [5]. Alternatively, syntenin interacts with the target protein and forms a functional complex on the membrane [10,15].

The functional diversity of syntenin is based on its binding partner-dependent roles, and syntenin can regulate membrane architecture through its binding partners. Here, we review the properties of syntenin in regulating neuronal morphogenesis, tumor metastasis and exosome biogenesis through regulating the subcellular trafficking of its interacting partners.

## 2. Structure and Regulation of Syntenin

Syntenin was first identified as a binding partner of the C-terminal cytoplasmic domain of syndecan, one of the heparan sulfate proteoglycans, via a yeast two-hybrid system [1]. At almost the same time, Lin et al. identified a series of genes whose expression levels are controlled by treatment with a combination of interferon-γ and mezerein in human melanoma cells. One of these genes was *mda-9*, encoded syntenin [2]. Syntenin is composed of 298 amino acid residues in humans, and its sequence is highly conserved across species from vertebrates to arthropods. Syntenin consists of four domains: an N-terminal domain, the first PDZ domain, the second PDZ domain, and a C-terminal domain (Figure 1A). In general, the PDZ domain binds to the C-terminal peptide of the partner protein at the plasma membrane and intracellular membranes [16,17].

The N-terminal domain of syntenin contains an autoinhibitory domain, possibly regulated by tyrosine phosphorylation of the N-terminal domain [19]. However, the distinct function of the C-terminal domain has not been clarified, only its influence on the structure and stability of the syntenin protein [20]. A recent study suggests that the C-terminal region of syntenin is required for its homodimerization, which is important for syndecan-related signaling [21].

PDZ domains are classified into three groups depending on their target peptide sequence. The three C-terminal residues of the partner protein determine the class—class I (-S/T-X-Φ), class II (-Φ-X-Φ), and class III (-D/E-X-Φ) (Φ represents a hydrophobic residue and X represents any amino acid residue)—and syntenin has been shown to bind the three groups of proteins with low to moderate affinity [22,23]. It is known that the PDZ1 and PDZ2 domains of syntenin have different preferences for binding partners; PDZ1 binds peptides of classes I (e.g., interleukin-5 receptor α chain (IL-5Rα) [22]) and III (e.g., CD63 [24] and Merlin [22]), while PDZ2 interacts with classes I (e.g., IL-5Rα [22] and neurofascin [16]) and II (e.g., syndecan [16,22], ephrin B [25], syntaxin1A [7], and neurexin [16]). However, PDZ1 has shown to interact with neurexin (class II) [26], and PDZ2 can interact with c-Src (class III) [27]. In addition, the binding properties of PDZ1 and PDZ2 are different; PDZ1 shows weak binding to its target proteins—that is, most peptide ligands preferentially bind the PDZ2 domain of syntenin [22] (Figure 1A). Syntenin-interacting partners are summarized in Table 1.

The PDZ domain consists of 80–90 amino acid residues that form globular domains. The common structure of PDZ domains comprises six β strands (β1–β6) and two α helices (α1 and α2). The C-terminal peptides of partner proteins are inserted into a groove between the β2 strand and the α2 helix of the PDZ domain as an antiparallel β-strand against β2 [28]. The three-dimensional structure of syntenin has already been investigated by crystal structural analysis of syntenin alone [21,22] and of syntenin with the C-terminal peptides of the binding partner, such as syndecan and IL-5Rα (Figure 1B) [18,23]. How the interacting groove accepts the side chains of the C-terminal residues of the binding partner has been well-studied. PDZ1 and PDZ2 domains of syntenin have less than 30% amino acid identity; however, the crystal structures showed two domains that are structurally similar [22], except for two differences. One is that the length of the β2–β3 loop in PDZ2 is shorter than that in PDZ1. The other is that the target binding groove of PDZ1 is narrower than that of PDZ2 or of other common PDZ domains. This could explain the weaker binding property of PDZ1 to target proteins than that of PDZ2 [22].

On the other hand, both PDZ domains of syntenin have been shown to be required to interact with syndecan [1,16,26]. Structural analysis suggests that the two PDZ domains work cooperatively to interact with syndecan, indicating that tandem PDZ domains of syntenin play crucial roles in the formation of structural and functional supramodules [23]. Moreover, the cytoplasmic domain of syndecan can dimerize in the central region and the dimerized syndecans are ideally suited to being inserted into the two binding grooves of syntenin [23]. Such cooperative interaction of PDZ domains of syntenin will provide high affinity to dimerized syndecans.

Calcium/calmodulin-dependent serine protein kinase (CASK) is another PDZ protein that interacts with the C-terminal motif of syndecan, and is important for the organization of the postsynaptic membrane in neurons, as described later [29,30]. CASK is a member of the membrane-associated guanylate kinase homologs (MAGUK) protein family, harboring a single PDZ domain [31]. This PDZ domain shows significant binding to syndecan by itself, in contrast to the PDZ domain of syntenin [26]. Furthermore, CASK PDZ prefers to interact with syndecan-2 and -4, whereas full-length syntenin equally binds to all syndecan family proteins [26]. Syntenin and CASK compete to interact with syndecan and one inhibits the interaction of the other. An appropriate balance of CASK and syntenin may be important for proper synapse formation [10]. The PDZ domain proteins mentioned in this review are summarized in Figure 1C.

## 3. Regulation of Cellular Functions by Syntenin

Syntenin is present in various tissues, but recent studies show that syntenin expression is higher in neurons and some tumor cells than in other types of cells. High expression of syntenin was first found in metastatic melanoma [32] and, more recently, in breast cancer [33,34], suggesting that syntenin could be involved in highly metastatic tumors, most likely by changing membrane properties. However, syntenin is also moderately expressed in nondividing neurons. Neurons are connected via synapses, which are unique structures of membranes supporting neurotransmission. Thus, syntenin can be involved in the regulatory mechanisms of highly motile membrane structures in neurons and invading tumor cells. In this section, we consider the roles of syntenin in membrane dynamics in neurons and tumor cells. Additionally, we discuss syntenin-mediated exosome biogenesis, which was recently clarified.

### 3.1. Regulation of Synapses

Syntenin was isolated as a binding partner of syndecan, one of the synaptic proteins that regulates synapse formation. Neuronal synapses are asymmetrical cell-cell contact sites that have neurotransmitter release machinery in the presynaptic terminals and multiple receptors and signaling molecules at the postsynaptic membrane. Numerous studies have shown that syntenin regulates the intracellular trafficking of binding proteins and controls downstream signaling molecules, resulting in the modulation of synaptic functions.

#### 3.1.1. The Main Effector Syndecan and Synapse Formation

Syndecan family proteins harbor heparan sulfate side chains on their extracellular domains. With their heparan sulfate modifications, syndecans act as coreceptors for growth factors, including fibroblast growth factors (FGF) [35]. Syndecans also function as adhesion molecules that control cell migration, cell-cell interactions, and cell-extracellular matrix interactions. Expression of syntenin mutants that are defective in binding PIP_2_ resulted in the accumulation of syndecan in recycling endosomes, indicating that syntenin is involved in syndecan recycling [36]. Interaction between syntenin and syndecan is necessary for the return of syndecan to the plasma membrane but not for the internalization of syndecan to the recycling endosome (Figure 2A) [36].

Postsynaptic localization of syndecan is important for the maturation of dendritic spines, which are protrusions from dendrites and receive excitatory inputs. In neuronal culture, syndecan levels increase with the days in culture in conjunction with synaptic maturation. Forced expression of syndecan in young cultured neurons induces the morphological maturation of dendritic spines [37]. The C-terminal EFYA sequence in syndecan is required for its interaction with syntenin. Overexpression of syndecan lacking the EFYA sequence failed to induce the morphological maturation of spines, which remain in an elongated, immature form [37]. Similar to syntenin, CASK also interacts with the C-terminal motif of syndecan-2 and promotes dendritic spine maturation [29,30]. Therefore, CASK-mediated postsynaptic localization of syndecan is crucial for spine maturation (Figure 2A). Phosphorylation of tyrosine in the EFYA sequence and Ser-183 (also in the C-terminal) of syndecan impeded the binding of syndecan to syntenin [38,39]. Posttranslational modifications of syndecans can control their intracellular trafficking. Thus, interaction between syntenin and syndecan may be important for the postsynaptic membrane localization of syndecan and for synapse maturation. However, it is suggested that increased amount of syntenin disrupts syndecan-mediated mature spine formation [10].

#### 3.1.2. The Regulator Rheb and Neurodevelopmental Disorders

Rheb is a small G protein enriched in the brain, and its expression is regulated by neuronal activity [40]. Syntenin binds to Rheb, and this interaction depends on the guanosine triphosphate (GTP)- or guanosine diphosphate (GDP)-bound state of Rheb [10]. Syntenin prefers to interact with GDP-bound Rheb and the Rheb/syntenin complex is susceptible to proteasomal degradation. This reduction in syntenin levels is important for normal spine maturation [10]. The intrinsic GTPase of Rheb is activated by Tsc1 and Tsc2, which are the gene products responsible for tuberous sclerosis complex (TSC). TSC is a neurodevelopmental disorder characterized by intractable epilepsy, intellectual disability and autism. *Tsc1* or *Tsc2* gene mutation forces Rheb into the GTP-bound state (active form), whereas the normal Tsc1-Tsc2 complex enhances the hydrolysis of GTP, leading Rheb into the GDP-bound state (inactive form) [41]. GDP-bound Rheb tightly associates with syntenin, and the Rheb/syntenin complex is prone to degradation by the proteasome (Figure 2A, left) [10]. However, GTP-bound Rheb loosely interacts with syntenin, which in turn bind to another protein such as syndecan-2 or ephrinB3. Usually, syndecan-2 binds with CASK, which stabilizes dendritic spines through association with the cytoskeleton via protein 4.1 [29]. Increases in syntenin levels exclude CASK from syndecan-2, and a large amount of syntenin could occupy syndecan-2 in neurons. This stoichiometric imbalance between CASK and syntenin could destabilize spine architecture (Figure 2A, right). We found that mutation in *Tsc2* causes filopodia formation along the dendrites in cultured neurons [42] and that knockdown of syntenin restored spine formation in *Tsc2^+/−^* neurons [10] (Figure 2B). Conversely, knockdown of syntenin in wild-type neurons resulted in the formation of immature spines [10]. Thus, altered syntenin levels disrupt proper spine formation in neurons.

Aberrant spine morphogenesis could cause functional impairment in synaptic plasticity. Thus, it is plausible that the mental retardation in TSC patients, caused by mutation of the *Tsc1/2* genes, depends on abnormal synaptic function via the imbalance between syntenin and CASK. To test this hypothesis, we investigated whether *Tsc2^+/−^*rats (Eker rats) had cognitive deficits with contextual fear discrimination tests, using a following protocol (Figure 2C, top). 1) On the first day, a training session was carried out. 2) On the next day, the conditioned fear response was assessed, and the percentage of freezing time during the test was compared. To evaluate context-dependent conditioned responses, trained rats were assessed in the same training context and in a novel context. These tests revealed that *Tsc2^+/−^*rats showed memory deficits. Wild-type rats with intracerebroventricular injection of scrambled small interfering RNA (siRNA) showed a significant difference in the freezing time between the training context and the novel context (*n* = 7, *p* = 0.027), whereas *Tsc2^+/−^*rats with scrambled siRNA injection failed to show a significant difference between these conditions (*n* = 10, *p* = 0.421) (Figure 2C, bottom). Knockdown of syntenin in the brain via chronic intracerebroventricular infusion of siRNA restored contextual fear discrimination in *Tsc2^+/−^*rats (*n* = 10, *p* < 0.001) (Figure 2C, bottom). Consistent with the data on dendritic spines, knockdown of brain syntenin caused memory impairment in wild-type rats (*n* = 7, *p* = 0.075) (Figure 2C, bottom). These results suggest that elevated or reduced expression levels of syntenin in animals caused cognitive deficits via abnormal spine morphogenesis.

Syntenin has been shown to interact with ephrin-B3 for excitatory shaft synapse formation [43,44]. Cultured neurons from *Tsc2^+/−^*rats showed an increased number of excitatory shaft synapses, and the formation of these synapses was ephrin-B3-dependent. Because knockdown of syntenin or ephrin-B3 reduced the number of shaft synapses in *Tsc2^+/−^*neurons [10], syntenin dissociated from Rheb may interact with ephrinB3 along the dendritic shaft to promote shaft synapse formation (Figure 2B).

Overall, Rheb/syntenin signaling can play pivotal roles in synapse formation. The stoichiometric balance between syntenin and CASK for association with syndecan is important for mature spine synapse formation. Loss of the balance between syntenin and CASK induces the formation of immature, thin spines and increases the number of dendritic shaft synapses (Figure 2A). Interestingly, *CASK* is one of the genes responsible for Ohtahara syndrome [45], which is accompanied by intractable epilepsy and intellectual disability in infancy [46]. Both of these symptoms are similar to those of TSC patients, suggesting that a syntenin/CASK imbalance could cause these neurodevelopmental disorders. Reduction of syntenin in the brain appears to be correlated with the restoration of proper spine formation and normal behavior in TSC model mice. Development of syntenin inhibitors could shed light on treatment for intractable epilepsy, intellectual disability, and autism.

#### 3.1.3. Association with Other Synaptic Proteins

Moreover, syntenin interacts with a variety of glutamate receptors [47,48,49,50]; however, its roles in synaptic transmission have not been elucidated. Previous reports suggest the role of syntenin in targeting interacting partners to subcellular locations such as synapses and regulating synaptic integrity by changing membrane architecture. Recent studies revealed that syntenin knockout mice showed impairment in the extinction of cued fear memory [51], suggesting that synaptic function controlling memory retention may be regulated by syntenin, though precise molecular or cellular mechanism is still unknown.

### 3.2. Tumor Cell Regulation

Syntenin is also known as melanoma differentiation-associated gene-9, or mda-9, the expression of which is upregulated by interferon treatment [2]. Subsequent studies revealed that syntenin is involved mainly in the metastasis of various types of tumor cells, including melanoma [27,52,53], urothelial cell carcinoma (UCC) [54], breast cancer [34,55], small cell lung cancer [56], and glioma cells [14,57]. Here, we show how syntenin mediates the malignant progression of tumor cells, especially in metastasis, by regulating cell membrane motility.

#### 3.2.1. Tumor Cell Invasion

Syntenin has been found to be highly expressed in invasive and metastatic cell lines [14,58]. Overexpression of syntenin in cells induced increased migration and invasion [52,59]. Conversely, knockdown of syntenin resulted in a decrease in the invasive ability of glioblastoma, UCC, and breast cancer cells in vitro [14,54,60]. In addition, xenograft experiments revealed that expression of small hairpin RNA (shRNA) targeting syntenin induces massive reductions in invasion, tumorigenesis, and angiogenesis in vivo [14,60]. Knockdown of syntenin improved the survival rate after xenografting of glioblastoma cells [14]. Thus, tumor cells invade by upregulating syntenin expression.

#### 3.2.2. Mechanism Acting through Focal Adhesion Kinase and c-Src

Precise mechanisms of syntenin-mediated cell migration remain elusive. Several studies have indicated that syntenin-interacting partners are involved in cell migration and cytoskeletal rearrangement. Syntenin binds to cellular Src kinase (c-Src) and this interaction results in its activation of the downstream signaling, such as the nuclear factor-kappa B (NF-κB) pathway [27,61]. c-Src has pivotal roles in cell motility, invasiveness, and malignant progression in a number of cancers [62]. Syntenin co-localizes with c-Src in metastatic melanoma and glioma, especially in areas of focal adhesion [53]. c-Src also binds to phosphorylated focal adhesion kinase (FAK) (Figure 3A). The interaction of FAK and Src kinase leads to the coordination of signaling through multiple pathways that influence the regulation of cell migration, tumor growth, and invasion [63,64]. Syntenin expression levels were correlated with the formation of the FAK-Src complex and active FAK levels in melanoma cells [52]. Phospho-FAK levels were decreased by knockdown of syntenin, and dominant-negative FAK expression significantly reduced syntenin-dependent migration of melanoma cells in vitro [52].

FAK-Src signaling activates the NF-κB pathway, which is involved in invasion-related transcriptional activity. NF-κB normally binds to IκBα and is inactivated. When upstream signaling is activated, IκB kinase (IKK)-mediated phosphorylation of IκBα results in its degradation (Figure 3A). Consequently, free NF-κB translocates to the nucleus, where it binds to target DNA sequences and controls gene expression [61]. NF-κB regulates the expression of genes involved in cell motility and invasion.

*MMP 2* is a target gene of NF-κB [52] and is one of the key players in tumor cell metastasis by degrading the extracellular matrix [65]. Syntenin signaling lead to increases in MMP2 expression in melanoma and glioblastoma [14,52], indicating that syntenin can induce tumor cell invasion through the expression of matrix-metalloprotease (MMP) 2 (Figure 3A).

Another study suggested that NF-κB regulates the expression of RhoA and Cdc42, small G proteins regulating actin cytoskeleton rearrangement [60]. Knockdown of syntenin in breast cancer cell lines resulted in a decrease in the levels of active RhoA and Cdc42. Reduced syntenin expression conversely altered cell morphology and the actin cytoskeleton (Figure 3A). Cells with high syntenin expression show lamellipodia- and filopodia-rich morphology and massive actin fibers, whereas low expression of syntenin induces a rounded cell shape and low abundance of actin fibers in the cells (Figure 3B) [60]. This regulation of actin fibers by syntenin/NF-κB signaling could be involved in the metastasis observed in xenograft experiments. Decreased expression of syntenin resulted in poor invasiveness in vivo, whereas NF-κB overexpression restored aggressive invasion even with syntenin knockdown [60].

Conclusively, these observations indicate that syntenin interacts with c-Src and regulates its activity, leading to the expression of invasion-related genes through the activation of NF-κB. Because c-Src regulates other proteins involved in cell motility, cytoskeletal rearrangement and membrane architecture [66,67], the syntenin/c-Src interaction might enhance cancer metastasis through activation of other pathways.

#### 3.2.3. Another Mechanism Acting through Activated Leukocyte Cell Adhesion Molecule and Merlin

Syntenin is involved in the regulation of cell motility by interacting with other proteins. Activated leukocyte cell adhesion molecule (ALCAM) is a binding partner of syntenin. ALCAM is a transmembrane protein that interacts with ezrin, an F-actin binding protein. Thus, the ALCAM/syntenin/ezrin complex is responsible for recruiting F-actin to a particular adhesion spot on the plasma membrane and for regulating cell adhesion strength (Figure 3C) [15]. Because ALCAM has been implicated in the aggressiveness of breast cancer [68,69], colorectal cancer [70,71], lung cancer [72], melanoma [73,74], and glioblastoma [75], the ALCAM/syntenin interaction might regulate cell motility through controlling adhesion strength.

Merlin, an four-point-one, ezrin, radixin, moesin FERM domain-containing actin-binding protein, is also a syntenin-interacting protein. Merlin is encoded by the *NF2* gene, mutations in which cause type 2 neurofibromatosis in humans [76]. When syntenin was overexpressed, merlin localized below the plasma membrane with some signals in filopodia extensions in HeLa cells. Conversely, cells with downregulation of syntenin exhibited a flattened morphology, and more importantly, merlin showed granular localization within the cytoplasm with decreased signal below the plasma membrane [76]. These results suggest that syntenin-mediated merlin regulation controls actin fiber rearrangement and cell motility in tumor metastasis.

#### 3.2.4. Syntenin as a Target for Anticancer Drugs

Because syntenin is involved in several aspects of malignant progression in cancer cells, it could be a therapeutic target [77,78]. Syntenin has been shown to be involved in the radiosensitivity of glioma cells. Patients with tumors with higher expression of syntenin showed significantly shorter survival times after radiation therapy than other glioma patients [77]. Knockdown of syntenin resulted in reduced viability of glioblastoma multiforme (GBM) cells in vitro after irradiation. A novel pharmacological inhibitor of syntenin, PDZ1i, interacts with the PDZ1 domain of syntenin and is expected to disrupt the interaction between syntenin and its binding partners [77]. PDZ1i effectively reduced tumor invasion in an animal xenograft model of GBM, and combined therapy with radiation and PDZi administration extended survival and GBM invasion in vivo [77]. Development of PDZ2 inhibitors would be anticipated in the near future.

### 3.3. Regulation of Exosome Biogenesis

Exosomes are small secreted vesicles that play key roles in intercellular communication, including neuron-neuron, neuron-glia, and glia-glia crosstalk [79,80,81,82]. In addition, exosome-mediated cell-cell communication is involved in tumor metastasis, cell proliferation and angiogenesis [83,84]. Syntenin interacts with the proteins involved in the biogenesis of exosomes, indicating that syntenin regulates intracellular membrane rearrangement and the cargo of these exosomes.

#### 3.3.1. Interaction with CD63 and ALG-2 interacting protein X

Tetraspanin, a member of the transmembrane protein family, has four transmembrane α-helices and two extracellular domains. It was previously shown that tetraspanins are abundant on exosomes [85]. Syntenin was coimmunoprecipitated with various tetraspanins (e.g., CD9, CD63, and CD81) from cell lysates, and CD63 showed a direct interaction with syntenin [24]. The PDZ and C-terminal domains of syntenin were required for interaction with the C-terminal domain of CD63, and this interaction may recruit syntenin to the plasma membrane [24]. CD63-syntenin complex controls membrane trafficking from early endosome to multivesicular endosomes.

ALG-2 interacting protein X (ALIX) is another interacting partner of syntenin that is involved in exosome biogenesis [12]. ALIX interacts with several endosomal sorting complex required for transport (ESCRT) proteins, participating in the membrane budding and neck abscission processes [86]. Importantly, ALIX also interacted with LYPX(n)L sequences on retroviral membrane proteins, allowing enveloped viruses to replace the original cellular mechanism of membrane-domain biogenesis and exit cells [87]. Coincidentally, three repeats of the LYPX(n)L sequence are found in the N-terminal domain of syntenin, and ALIX can bind the N-terminal domain of syntenin [12].

CD63 has been shown to bind to a capsid protein of human papillomaviruses (HPV). The CD63-syntenin complex mediates HPV trafficking steps to multivesicular bodies in HPV infection (Figure 3D) [88]. ALIX also participates in the HPV infection. Syntenin mutants lacking ALIX interaction ability were unable to promote early steps of viral infection [88]. Thus, the CD63-syntenin complex requires ALIX to control the vesicle trafficking mechanism in the HPV infection.

Furthermore, syntenin-ALIX complex mediates exosome biogenesis in multivesicular bodies. Syndecan, syntenin, and ALIX can form a tripartite complex [12], and co-accumulate in exosome vesicles. CD63 is also found in the same vesicle fraction (Figure 3E). Overexpression of syntenin increased the number of exosomes, and knockdown of syntenin reduced the number of vesicles. However, confocal fluorescence spectroscopic analysis revealed that the amount of exosomal cargo (GFP-tagged CD63) was not affected by the overexpression of syntenin [12]. Therefore, syntenin controls the number of exosomes released rather than the amount of cargo loaded in the modal exosome. These findings also suggest that the syndecan-syntenin pathway can account for the formation of a major proportion of the exosomes (Figure 3E).

#### 3.3.2. Regulation by c-Src

Interestingly, knockdown of Src, which is another syntenin-binding partner, also reduced exosome production, indicating that syntenin mediates the function of Src in exosome biogenesis [89]. Knockdown of Src failed to load syntenin into the endosome, the key step in the budding process at endosomal membranes. Src can phosphorylate the tyrosine residue on the membrane-proximal DEGSY motif on syndecan and Y46 on syntenin. Expression of the phosphomimetic form of syndecan or syntenin increased exosome production; however, Src inhibitor treatment decreased endosomal budding when either syndecan or syntenin was wild-type, but not when both syndecan and syntenin were phosphomimetic [89]. These results indicate that Src-mediated phosphorylation of tyrosine residues on both syndecan and syntenin is necessary for exosome biogenesis (Figure 3E). A recent study also revealed that Src binds to ALIX to promote the biogenesis of exosome (in which active Src is encapsulated) [90]. Thus, Src can be a pivotal regulator for syndecan-syntenin-ALIX mediated exosome biogenesis.

#### 3.3.3. Role in the Parkinson’s Disease

A recent study indicated that exosome-associated proteins, including syntenin, were more abundant in exosomes from Parkinson’s disease patients than in exosomes from the healthy subject group [91]. These changes in protein composition were detected despite similar exosome numbers across the groups, suggesting that they would reflect exosome subpopulations with distinct functions. Furthermore, exosomes from patients with Parkinson’s disease conveyed neuroprotective effects [91]. Syntenin could be involved in this neuroprotective exosome release. These studies demonstrate another role of syntenin activity in membrane-associated structures to promote both cell-cell communication in the central nervous system and tumor pathogenesis.

### 3.4. Syntenin in other Biological Processes

Syntenin participates in biological processes other than those mentioned above, including synaptic regulation, tumor metastasis, and exosome biogenesis. Through the regulation of membrane trafficking, syntenin is involved in the neurite outgrowth and induction of anoikis-resistance of glioma stem cells (GSC). We will briefly describe how syntenin is involved in these biological processes.

#### 3.4.1. Neurite Outgrowth

Uncoordinated (Unc)51.1/Ulk1 and Rab5 are interacting partners of syntenin [11]. Unc51.1 belongs to the subfamily of Ser/Thr kinase, involved in axonal elongation in cerebellar granule cells and isolation of the membrane at the autophagosome [92]. Rab5 is a small G-protein, regulating the early endocytic membrane [93,94]. Unc51.1 and a synaptic GTPase-activating protein (SynGAP) [95,96] co-localized within the axon of granule cells. As SynGAP stimulates Rab5 GTPase and reduces its activity, close proximity of the Unc51.1/SynGAP complex to Rab5 may change the membrane organization in the axons [11]. Syntenin acts as a scaffold protein for Unc51.1/SynGAP and Rab5 to elongate axons through the inhibition of endocytosis (Figure 3F) [11].

#### 3.4.2. Anoikis-Resistance of Glioma Stem Cells

Syntenin is involved in the self-renewing of glioma “stem cells.” The concept of the “cancer stem cell” is that cancer is comprised of a variety of cells with the potential to metastasize, interact with the stroma, and regrow after therapy [97,98,99]. The expression level of syntenin was higher in GSC than in non-stem glioma cells [13]. Syntenin activates a PKCα-dependent phosphorylation of the anti-apoptotic Bcl-2 protein, which is crucial for protective autophagy and anoikis-resistance in GSC (Figure 3G) [100]. In this mechanism, FAK activity is also required for syntenin-mediated PKC signaling.

## 4. Concluding Remarks

Syntenin shows a surprising diversity of interacting partners, suggesting that it plays a number of flexible cell type-specific roles. Syntenin forms a variety of complexes. Some of these complexes are specific to a particular cell type, and others are specific to a subcellular compartment involved in numerous intracellular signaling cascades. Because syntenin has two PDZ domains, it can interact simultaneously with two different partners. For instance, syntenin works as a scaffold protein dependent on the intracellular environment or in the subcellular compartment by regulating membrane architecture rearrangement.

Syntenin interacts with other synaptic proteins, including ephrin family proteins [26,43], neurexin [26], and syntaxin [7], all of which guarantee precise synapse function and localize between the pre- and postsynapse. The molecular mechanism by which syntenin regulates these proteins has not been elucidated. Because syntenin can regulate the subcellular trafficking of various proteins, it might control the internalization or insertion of these binding partners in the synaptic membrane. Protein endocytosis and protein integration in the synaptic membrane is a pivotal event in synaptic plasticity. Thus, syntenin can be one of the key regulators of neuronal synapse formation.

As noted, syntenin has frequently been identified as indispensable for cell invasion and metastasis in various cancers [14,60], indicating that syntenin plays a pivotal role in the signaling controlling cell motility. It might be possible to develop novel cancer therapeutics that target syntenin. As evidenced by the results of decades of research on cancer drug delivery and development, a single compound that can completely eliminate malignant tumors is unlikely. Nevertheless, molecules targeting syntenin could serve as a new strategy to augment conventional therapeutics. Therefore, targeting syntenin could be a part of a combination approach with other chemotherapy or radiotherapy. In this context, inhibitors of syntenin will provide a novel strategy for effectively treating and potentially preventing the malignant progression of tumors.

Exosome biogenesis is involved in cell-cell communication in both central nervous system function and tumor development. Because syntenin is involved in exosome biogenesis, synaptic plasticity and tumor metastasis could also be affected by another function of syntenin. Regulation of syntenin in exosome biogenesis, such as interaction with ALIX, can provide another aspect to inhibit the exosome-mediated tumor development. Moreover, exosome biogenesis in neuronal development and pathogenesis still remains elusive. Further analysis of syntenin in exosomal communications in the brain will pave roads leading to insight into proper nervous system formation and will lead to the development of new diagnostic methods for neurodevelopmental and neurodegenerative diseases.

## Figures and Tables

**Figure 1 ijms-20-04171-f001:**
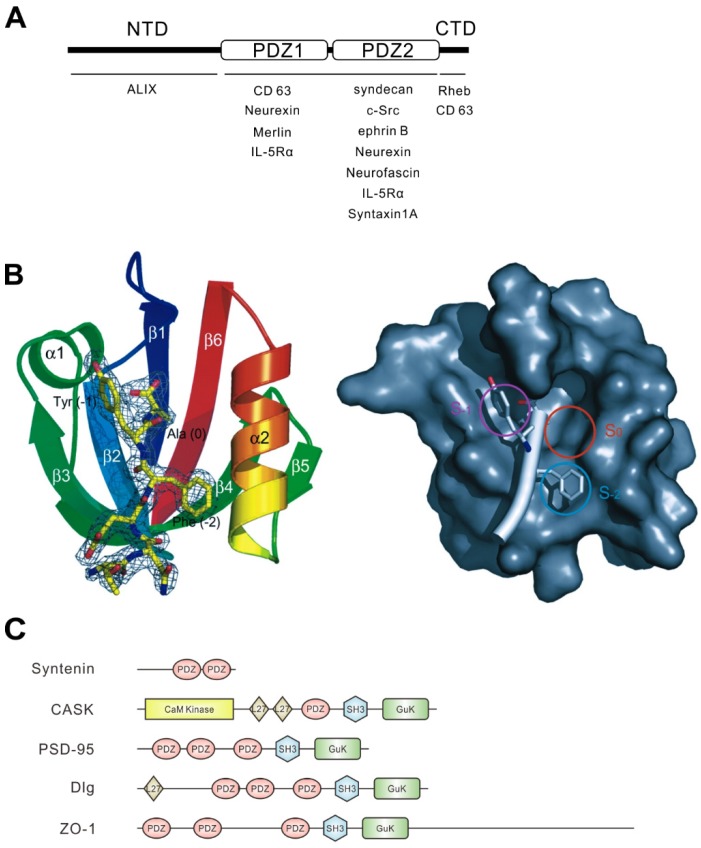
Structural character of syntenin and PDZ domain. (**A**) Schematic illustration of the domain structure of syntenin. The N-terminal domain (NTD), the PDZ domain tandem repeat, and the C-terminal domain (CTD) of syntenin are indicated. The names of representative syntenin-binding proteins are shown beneath the domains required for their interaction. Neurexin showed controversial results, preferentially binding with PDZ1 or PDZ2, and CD63 requires both the CTD and PDZ1 for interaction. IL-5Rα can interact with both PDZ1 and PDZ2. (**B**) Ribbon diagram of the syntenin PDZ2 domain bound to the C-terminal peptide of syndecan-4 (TNEFYA) (yellow line diagram) (left). Surface representation of the syntenin PDZ2 domain showing three hydrophobic pockets and the syndecan-4 peptide (white bar) (right). Circles indicate three binding pockets. The three C-terminal residues are shown in C_α_ race. The side chains of tyrosine (−1) and phenylalanine (−2) occupy the two pockets S_−1_ and S_−2_, and alanine (0) occupies S_0_. Images and the legend are from Kang et al. [18]. (**C**) Domain organizations of PDZ proteins mentioned in this review. PDZ domains are shown in orange ellipses. Other domains are indicated: CaM Kinase, calmodulin-dependent kinase (CaMK)-like domain; GuK, guanylate kinase-like domain; L27, domain initially found in LIN2 and LIN7; SH3, Src homology 3 domain.

**Figure 2 ijms-20-04171-f002:**
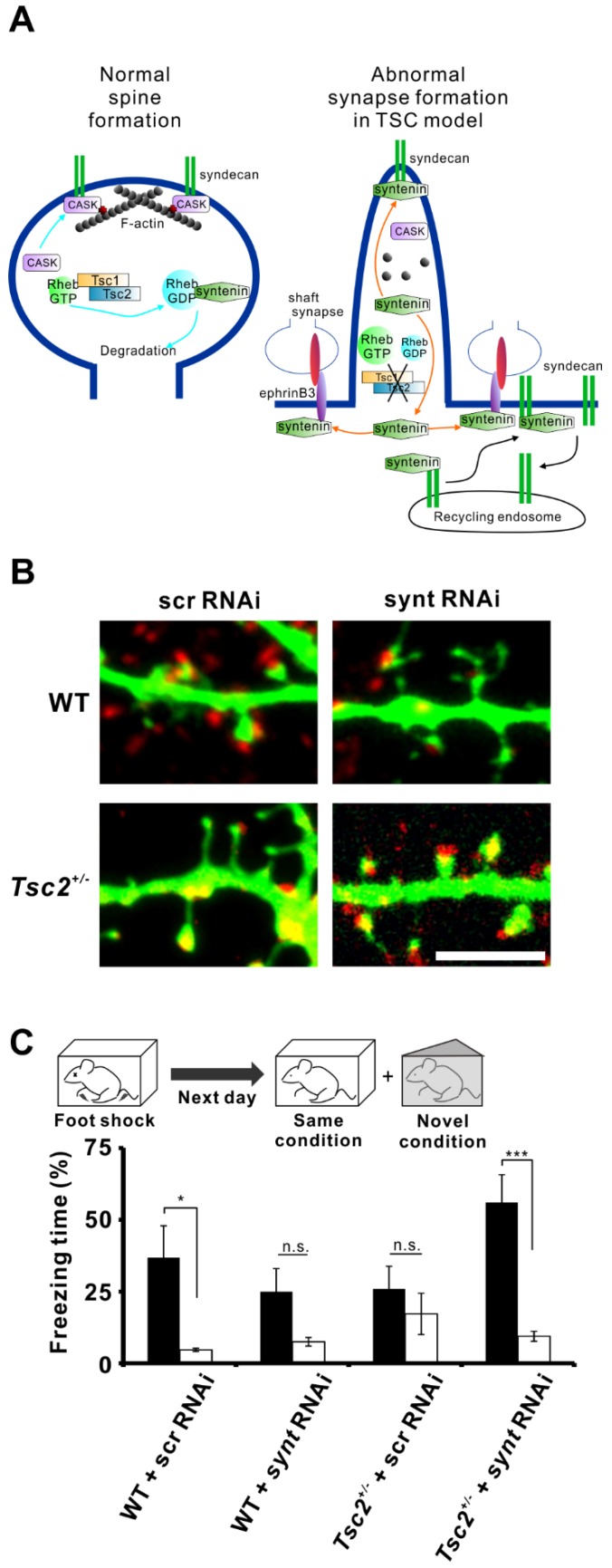
Syntenin is involved in dendritic spine morphology and memory impairment in tuberous sclerosis complex (TSC) model rats (**A**) Model depicting the role of syntenin in spine and shaft synapse formation. In wild-type neurons, Rheb-guanosine diphosphate (GDP) interacts with syntenin, promoting syntenin degradation. This decrease in syntenin levels facilitates the association of CASK with syndecan, leading to normal spine development (left). However, *Tsc* gene mutation increases the level of Rheb-guanosine triphosphate (GTP), which rescues syntenin from degradation. The accumulated syntenin inhibits the binding of CASK to syndecan and associates with ephrinB3, resulting in impaired spine formation and enhanced shaft synapse formation (right). Syntenin also mediates the translocation of syndecan from the recycling endosome to the cell surface (black arrows). (**B**) Representative images of dendrites of cultured rat wild-type and *Tsc2^+/−^* neurons transfected with an enhanced green fluorescent protein (EGFP) plasmid and with syntenin small interfering RNA (siRNA) or scrambled siRNA. Elongated spines in *Tsc2^+/−^* neurons are restored to normal morphology by the knockdown of syntenin. The neurons were immunolabelled with anti-vGlut1 (red). Scale bar, 5 μm. (**C**) *Tsc2^+/−^* rats (Eker rats) showed contextual memory deficits, but knockdown of syntenin expression in the brain reversed their memory impairment. Schematic illustration of the contextual fear discrimination test (top). Quantification of the freezing time in each condition is shown (bottom).

**Figure 3 ijms-20-04171-f003:**
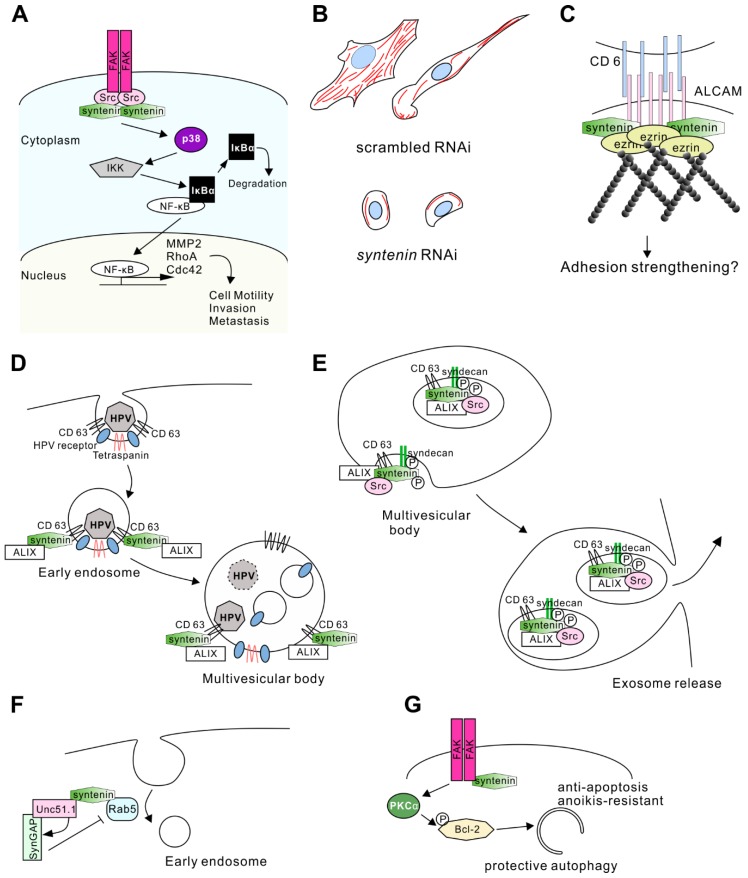
Syntenin-mediated signaling in the regulation of membrane structure and cell motility. (**A**) Syntenin facilitates the motility and metastasis of tumor cells. Syntenin activates the focal adhesion kinase (FAK)-c-Src kinase complex and induces the degradation of Inhibitor kappa B Iκ-B, resulting in the translocation of nuclear factor-kappa B (NF-κB) into the nucleus. Consequently, NF-κB induces the gene expression of matrix-metalloprotease (MMP)-2, ras homolog gene family, member A (RhoA), and cell division control protein (Cdc)42, inducing the degradation of the extracellular matrix and increasing cell motility by F-actin regulation. (**B**) Schematic illustration of representative cell shapes and F-actin rearrangement in breast cancer cells upon syntenin RNA interference (RNAi) treatment. The images refer to Menezes et al. [60]. (**C**) Syntenin binds to the cytoplasmic tail of activated leukocyte cell adhesion molecule (ALCAM) and strengthens the binding by coupling ALCAM to the actin cortex. Ezrin also connects some of the ALCAM molecules to actin at juxtamembrane sites. Upon binding of ALCAM to its ligand cluster of differentiation (CD)6, the complex is strengthened, and syntenin as well as ezrin can be phosphorylated, leading to outside-in signaling and ultimately to cellular responses, most likely adhesion strengthening. (**D**) Model of the CD63-syntenin-ALIX-dependent trafficking of human papillomaviruses (HPV). Its endocytosis occurs at the CD63-enriched microdomains of plasma membrane. Virus uptake starts with recruitment of syntenin and ALG-2 interacting protein X (ALIX) at early endosomes. The CD63-syntenin-ALIX complex promotes post-endocytic trafficking of HPV to multivesicular bodies. (**E**) Syntenin-ALIX complex is involved in exosome biogenesis and release. Syntenin associates with syndecan and CD63 in exosomes, and promotes their releases. Biogenesis is enhanced by interaction between ALIX and Src, which phosphorylates syndecan and syntenin. (**F**) Syntenin works as an adaptor protein for Unc51.1, SynGAP, and Rab5. Unc51.1 activates SynGAP and downregulates Rab5 activity, resulting in inhibition of the early endosome pathway and the promotion of neurite extension. (**G**) Syntenin-mediates anoikis-resistance. Syntenin can activate PKCα in an FAK-dependent manner, and induce Bcl-2 phosphorylation. This signaling promotes anti-apoptotic protective autophagy instead of toxic autophagy.

**Table 1 ijms-20-04171-t001:** Overview of interaction partners of syntenin mentioned in this review.

Binding Partner	Where to Interact with Syntenin	Reference
ALG-2 interacting protein X (ALIX)	NTD	[12]
Cluster of differentiation (CD)63	PDZ1 and CTD	[24]
Neurexin	PDZ1 and PDZ2	[16,26]
Merlin	PDZ1	[22]
IL-5Rα	PDZ1 and PDZ2	[22]
Syndecan	PDZ2	[16,22]
cellular Src kinase (c-Src)	PDZ2	[27]
Ephrin B	PDZ2	[25]
Neurofascin	PDZ2	[16]
Syntaxin 1A	PDZ2	[7]
Ras homolog enriched in brain (Rheb)	CTD	[10]
Activated leukocyte cell adhesion molecule (ALCAM)	Not examined. ALCAM has PDZ binding motif at its C-terminal (-TEA, class I)	[15]
Uncoordinated (Unc)51.1	PDZ1 and PDZ2 ?	[11]
Ras-associated binding (Rab)5	PDZ1 and PDZ2 ?	[11]

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
