# Peer review of "Syntenin: PDZ Protein Regulating Signaling Pathways and Cellular Functions"

_ijms, 2019, doi:10.3390/ijms20174171_

Round 1
Reviewer 1 Report
This review focuses on the interacting partners of syntenin, which is a cell surface signaling and trafficking proteins. The functional roles of the interactions between syntenin and various binding parterners have been discussed in the context of synaptic membrane regulation, cancer metastasis and exosome biogenesis. The manuscript is very well written and informative and I recommend it to be published after minor revision.
To improve the clarity and readability of the manuscript, I have the following specific suggestions:
#1: Many molecular mechanisms mentioned in the manuscript are speculative, as indicated by the words “maybe”, “possible”, etc. The authors should provide some discussions on the future studies that could clarify these mechanisms to give readers a sense about where the field is going. Are there technical challenges that hamper the efforts on elucidation of the mechanisms. Any conflicting evidences, if existing, should be presented for the readers to understand whether there are any alternative mechanisms other than those discussed in the manuscript.
#2: Line 139-140: The author should clearly explain the functional outcome of the binding of syntenin to GTP- and GDP-bound Rheb before describe the function/mechanisms of action of Rheb in detail. This would enhance the readability of this paragraph for readers who are not familiar with the functional role of syntenin in the context of its interaction with Rheb.
#3: Line 204-205: To enhance readability, a brief description of the interaction between c-Src and syntenin, and the downstream signal events (such as activation of NF-kappaB pathway) should be included here before the detailed discussion.
#4: Line 361: reference is needed to support the statement that “syntenin has … as indispensable for …”
Author Response
Reviewer#1
#1: Many molecular mechanisms mentioned in the manuscript are speculative, as indicated by the words “maybe”, “possible”, etc. The authors should provide some discussions on the future studies that could clarify these mechanisms to give readers a sense about where the field is going. Are there technical challenges that hamper the efforts on elucidation of the mechanisms. Any conflicting evidences, if existing, should be presented for the readers to understand whether there are any alternative mechanisms other than those discussed in the manuscript.
Regarding the mechanisms whose evidence is clear enough, without conflicting findings, we have removed “may”, “maybe”, and “possible” from the corresponding paragraphs. When the evidence is not solid or still predictive or estimative, we have replaced them with other words or described conflicting evidence. In addition, we have discussed future studies in the manuscript (Lines 233 – 236, 460 – 466).
#2: Line 139-140: The author should clearly explain the functional outcome of the binding of syntenin to GTP- and GDP-bound Rheb before describe the function/mechanisms of action of Rheb in detail. This would enhance the readability of this paragraph for readers who are not familiar with the functional role of syntenin in the context of its interaction with Rheb.
According to the reviewer’s suggestion, we have added a brief description on the functional outcome of the binding between syntenin and GTP- or GDP-bound Rheb (Lines 183 – 185).
#3: Line 204-205: To enhance readability, a brief description of the interaction between c-Src and syntenin, and the downstream signal events (such as activation of NF-kappaB pathway) should be included here before the detailed discussin.
According to the reviewer’s suggestion, we have added a brief description about the interaction between c-Src and syntenin, and downstream signal events (Lines 263 – 264).
#4: Line 361: reference is needed to support the statement that “syntenin has … as indispensable for …”
According to the reviewer’s suggestion, we have added references for this sentence (Line 450).
Reviewer 2 Report
Overall the review provides a single location for a wide range of information and references for syntenin, in particular the interactions of syntenin with other proteins. However, the review does not touch significantly on membrane dynamics as described in the title. The review is broad, but not deep. The manuscript could be aided by revision with an effort on focusing more tightly on regulation of membrane dynamics, a topic only superficially covered at this stage.
Additionally, there are a few shortcomings that should be addressed in a subsequent version of the manuscript:
1. The introduction is a bit generic and a bit more description of how syntenin functions as an adapter could be very helpful to the reader.
2. At the first instance, on line 23, PDZ should be defined.
3. The "structure and regulation of syntenin" section includes very little structural information. What about the three dimensional structure of syntenin, or homologous proteins with PDZ domains? A discussion of the structure of syntenin, or domains within, and the consequences of the structure would be helpful. Currently this section is more of a discussion of function or interactions.
4. The legend for figure 1 is long and ungainly. It contains much more discussion than is typical for a figure. Perhaps some of this discussion could be moved to the main text?
5. Much of the main text focuses on interactions or pathways, but does not give a lot of detail about membrane interactions or regulation of membrane dynamics. The discussion tends to be a higher level organismal view, much more so than one would expect from the current title of the manuscript.
Author Response
Reviewer#2
Overall the review provides a single location for a wide range of information and references for syntenin, in particular the interactions of syntenin with other proteins. However, the review does not touch significantly on membrane dynamics as described in the title. The review is broad, but not deep. The manuscript could be aided by revision with an effort on focusing more tightly on regulation of membrane dynamics, a topic only superficially covered at this stage.
Based on the contents of this manuscript, we have changed the title to “Syntenin: a PDZ protein regulating signaling pathways and cellular functions.”
The introduction is a bit generic and a bit more description of how syntenin functions as an adapter could be very helpful to the reader.
According to the reviewer’s suggestion, we have added a description of how syntenin functions as an adaptor protein (Lines 36 – 39).
At the first instance, on line 23, PDZ should be defined.
According to the reviewer’s suggestion, we have added the definition of PDZ and the origin of its name (Lines 23 – 26).
The "structure and regulation of syntenin" section includes very little structural information. What about the three dimensional structure of syntenin, or homologous proteins with PDZ domains? A discussion of the structure of syntenin, or domains within, and the consequences of the structure would be helpful. Currently this section is more of a discussion of function or interactions.
According to the reviewer’s suggestion, we have added the paragraphs and figures to provide structural information on syntenin (Lines 72 – 100, 109 – 118, Figure 1B and 1C).
The legend for figure 1 is long and ungainly. It contains much more discussion than is typical for a figure. Perhaps some of this discussion could be moved to the main text?
According to the reviewer’s suggestion, we have shortened the legend for Figure 1 (New Figure 2) and made it more concise (Lines 165 – 179).
Much of the main text focuses on interactions or pathways, but does not give a lot of detail about membrane interactions or regulation of membrane dynamics. The discussion tends to be a higher level organismal view, much more so than one would expect from the current title of the manuscript.
According to the reviewer’s suggestion, we have changed the title of the manuscript to match the content.
Reviewer 3 Report
In this manuscript, Shimada et al. discusses about syntenin, a dual PDZ domain-containing protein involved in the regulation of cell membrane architecture. The authors first described the structural components as well as regulatory mechanisms of syntenin, and then provided a comprehensive summary of the functional roles of syntenin in multiple biological processes including synaptic regulation, tumor cell invasion and metastasis, and exosome biogenesis. This manuscript provided a concise and well-organized overview of syntenin and should be of high interest for researchers in the corresponding fields. Therefore, the reviewer would recommend the acceptance of the manuscript with a couple of minor changes requested.
1) It was mentioned in the manuscripts that syntenin interacts with a wide range of proteins, and in the following sections the authors listed a number of representative interacting partners of syntenin. It is recommended that the authors also sort and assemble these interacting partners into a table for ease of reading.
2) In Section 3 of the manuscript, the authors talked about the three major biological processes that syntenin is involved in. The reviewer is just curious whether syntenin is participating in other biological processes as well and how far the research has gone. If so could the authors provide a short section after Section 3 to briefly talk about the functional implications of syntenin in other biological processes?
Author Response
Reviewer#3
1) It was mentioned in the manuscripts that syntenin interacts with a wide range of proteins, and in the following sections the authors listed a number of representative interacting partners of syntenin. It is recommended that the authors also sort and assemble these interacting partners into a table for ease of reading.
According to the reviewer’s suggestion, we have added the table that includes the list of syntenin-interacting partners mentioned in the review (Table 1, Line 120).
2) In Section 3 of the manuscript, the authors talked about the three major biological processes that syntenin is involved in. The reviewer is just curious whether syntenin is participating in other biological processes as well and how far the research has gone. If so could the authors provide a short section after Section 3 to briefly talk about the functional implications of syntenin in other biological processes?
According to the reviewer’s suggestion, we have added the section on other biological processes that syntenin participates in. (Lines 318 – 323, 411 – 433, Figure 3F, G).
Round 2
Reviewer 2 Report
The manuscript is much improved, however the new paragraph beginning on line 233 is awkward. I have neer seen a review state that "We are going to further clarify this mechanism in vivo, using syntenin knockout mice." This sentence should be deleted and the remaining portion of the paragraph sould be combined with the paragraph above.
Author Response
The manuscript is much improved, however the new paragraph beginning on line 233 is awkward. I have neer seen a review state that "We are going to further clarify this mechanism in vivo, using syntenin knockout mice." This sentence should be deleted and the remaining portion of the paragraph sould be combined with the paragraph above.
According to the reviewer’s suggestion, we have deleted the sentence and combined remaining sentences of the paragraph with the previous paragraph (Lines 232 – 235).